

# Phylogeography of *Arenaria balearica* L. (Caryophyllaceae): evolutionary history of a disjunct endemic from the Western Mediterranean continental islands

Javier Bobo-Pinilla[1,2], Sara B. Barrios de León[1], Jaume Seguí Colomar[3], Giuseppe Fenu[4], Gianluigi Bacchetta[5], Julio Peñas de Giles[6] and María Montserrat Martínez-Ortega[1,2]

[1] Department of Botany, University of Salamanca, Salamanca, Spain
[2] Biobanco de ADN Vegetal, Banco Nacional de ADN, University of Salamanca, Salamanca, Spain
[3] Department of Terrestrial Ecology, Instituto Mediterráneo de Estudios Avanzados (IMEDEA), Esporles, Spain
[4] Dipartimento di Biologia Ambientale, University of Roma "La Sapienza", Roma, Italy
[5] Centro Conservazione Biodiversità (CCB), Dipartimento di Scienze della Vita e dell'Ambiente, University of Cagliari, Cagliari, Italy
[6] Department of Botany, University of Granada, Granada, Spain

Corresponding author
Javier Bobo-Pinilla,
javicastronuevo@usal.es

## ABSTRACT

Although it has been traditionally accepted that *Arenaria balearica* (Caryophyllaceae) could be a relict Tertiary plant species, this has never been experimentally tested. Nor have the palaeohistorical reasons underlying the highly fragmented distribution of the species in the Western Mediterranean region been investigated. We have analysed AFLP data (213) and plastid DNA sequences (226) from a total of 250 plants from 29 populations sampled throughout the entire distribution range of the species in Majorca, Corsica, Sardinia, and the Tuscan Archipelago. The AFLP data analyses indicate very low geographic structure and population differentiation. Based on plastid DNA data, six alternative phylogeographic hypotheses were tested using Approximate Bayesian Computation (ABC). These analyses revealed ancient area fragmentation as the most probable scenario, which is in accordance with the star-like topology of the parsimony network that suggests a pattern of long term survival and subsequent *in situ* differentiation. Overall low levels of genetic diversity and plastid DNA variation were found, reflecting evolutionary stasis of a species preserved in locally long-term stable habitats.

## INTRODUCTION

Within the Mediterranean global biodiversity hotspot, the Tyrrhenian Islands represent ca. 22% of the total surface, and lodge a high percentage of endemic taxa (ca. 10-20%; *Contandriopoulos, 1990*; *Médail & Quézel, 1997*; *Bacchetta & Pontecorvo, 2005*; *Cañadas et al., 2014*). Some of these endemic plant species show narrow distributions (*Médail & Quézel, 1999*; *Thompson, 2005*; *Fenu et al., 2010*; *Bacchetta, Fenu & Mattana, 2012*),

but others are distributed in the major Western Mediterranean islands. Some endemic plant species shared by Corsica, Sardinia, and the Balearic Islands have been designated "Hercynian endemics" (*Mansion et al., 2008*) and are often considered palaeoendemic in the broad sense of the term (i.e., ancient or relict taxa often systematically isolated, *Favarger & Contandriopoulos, 1961*; *Greuter, 1995*; *Quézel, 1995*). The present distribution of such Hercynian endemic species has been attributed to the Oligocenic connections among the Western Mediterranean islands (*Greuter, 1995*; *Quézel, 1995*; *Thompson, 2005*), but this has not been tested in all cases. Additionally, the term "palaeoendemic" has been restricted in concept (*Thompson, 2005*) to include only clearly ancient isolated species in large genera (or monotypic genera) that usually show little variability. There are some endemic species showing distribution patterns that seem to be concordant with the geological history of the Western Mediterranean continental fragments, which have been commonly considered palaeoendemics. But, as it has not been yet demonstrated that they are of ancient origin and do not seem to be highly isolated within large genera, these do not fit into the restrictive concept of palaeoendemism proposed by *Thompson (2005)*. These species are referred to as disjunct endemics and *Arenaria balearica* L. from the family Caryophyllaceae is a good example.

The Mediterranean region has been affected by dramatic palaeogeographical events and by formidable bioclimatic changes during the Late Tertiary and Quaternary (*Kadereit & Comes, 2005*), which have influenced the structure and composition of the flora, have contributed to shape plant species distributions, and have modelled intraspecific genetic variability of species over the past million years (*Thompson, 2005*; *Médail & Diadema, 2009*).

Like most Western Mediterranean islands, Corsica, Sardinia, and Majorca are of the continental type and have been separated from each other by tectonic and glacio-eustatic processes (*Alvarez, 1972*; *Alvarez, Cocozza & Wezel, 1974*; *Rosenbaum, Lister & Duboz, 2002*; *Mansion et al., 2008*; *Mayol et al., 2012*). The post-Oligocene (which started ca. 30 Ma (million years ago)) progressive fragmentation of land masses previously constituting part of the Hercynian belt has been described elsewhere (*Alvarez, 1972*; *Alvarez, Cocozza & Wezel, 1974*; *Rosenbaum, Lister & Duboz, 2002*; *Speranza et al., 2002*; *Meulenkamp & Sissingh, 2003*; *Mansion et al., 2008*; *Salvo et al., 2010*).

The Tuscan Archipelago consists of seven small islands and several islets of different geological origins, which are also tectonic fragments that were once integrated within the Hercynian massif (*Salvo et al., 2010*). The granitic basement of Montecristo appears also to be partly a result of the volcanic activity displayed in the area over the past 10 Ma, giving rise as well to other volcanic islands in the region, such as Capraia (*Carmignani & Lazzarotto, 2004*).

With the closure of the Strait of Gibraltar (ca. 5.59 Ma; *Hsü, 1972*; *Garcia-Castellanos et al., 2009*) the Messinian Salinity Crisis of the Late Miocene started and some connections were established between North Africa, Corsica, Sardinia, and continental Europe, as well as between the Balearic Islands and Iberia; but no evidence of direct terrestrial corridors between Corsica or Sardinia and Balearic Islands have been documented (*Alvarez, 1972*; *Alvarez, Cocozza & Wezel, 1974*; *Rosenbaum, Lister & Duboz, 2002*; *Mansion et al., 2008*;

*Salvo et al., 2010*). During the Messinian, the Tuscan Archipelago may have connected Corsica, Sardinia, and the Italian Peninsula. The cycles of desiccation and transgression of the Mediterranean Sea in this period enabled interchanges of lineages of biota that predated the Messinian Salinity Crisis in all these territories (e.g., *Salvo et al., 2010*; *Molins et al., 2011*). The subsequent reopening of the Strait of Gibraltar (ca. 5.33 Ma; *Krijgsman et al., 1999*; *Garcia-Castellanos et al., 2009*) caused partial extinction and isolation of previously connected populations and seems to have promoted vicariant speciation and population divergence at least in some documented cases (e.g., *Quercus ilex* L. in *Lumaret et al., 2002*; *Anchusa crispa* Viv. in *Quilichini, Debussche & Thompson, 2004*; *Borago* L. in *Selvi, Coppi & Bigazzi, 2006*; *Abies* spp. in *Terrab et al., 2007*; *Anchusa* L. in *Bacchetta et al., 2008*; *Anchusa* L. in *Coppi, Mengoni & Selvi, 2008*; *Rodríguez-Sánchez et al., 2008*; *Salvo et al., 2008*; *Cephalaria* gr. *squamiflora* (Sieber) Greuter in *Rosselló et al., 2009*; *Bacchetta et al., 2012*; *Aquilegia* L. in *Garrido et al., 2012*).

The subsequent establishment of the Mediterranean climate (ca. 3–2 Ma) promoted the expansion of xerophytic elements and typically Mediterranean taxa (*Suc, 1984*; *Thompson, 2005*). Later, the cyclical climatic oscillations of the Quaternary Pleistocene (ca. 1.8–0.01 Ma) also significantly shaped the genetic structure and spatial distribution of the biota, leading to population differentiation and eventually to speciation (*Hewitt, 1999*). Particularly, during the Pleistocene glacial maxima the sea level was approximately 120–150 m lower than at present (*Yokohama et al., 2000*; *Church et al., 2001*; *Clark & Mix, 2002*; *Lambeck & Purcell, 2005*) and the Corsican and Sardinian coastlines were directly connected by land bridges (*Salvo et al., 2010*). These connections facilitated exchanges of plant species and have alternatively limited or favoured gene flow between populations of species distributed in both islands and probably also among them and the Tuscan islets (Fig. 1).

Several Mediterranean disjunct endemic species show high levels of morphological stability despite long-term isolation among populations distributed in different continental fragment islands (*Molins et al., 2011*, 3.2 Ma). The constancy of morphological characters over long time periods has frequently been related to low molecular evolutionary rates, although this may not be completely clear in all cases (*Casane & Laurenti, 2013*) and, recently, high levels of plastid DNA (cpDNA) diversity have been reported for the Tyrrhenian endemic *Thymus herba-barona* Loisel. (*Molins et al., 2011*). Also the apparent inconsistency between the fact that the Mediterranean region has undergone dramatic geological as well as climatic changes and the long persistence of Mediterranean endemic species has been explained as the result of reduced and isolated, but particularly stable, habitats (e.g., rocky habitats) suitable for species survival, within a sea of unsuitable landscapes (*Hampe & Petit, 2005*; *Thompson, 2005*; *Youssef et al., 2010*; *Molins et al., 2011*; *Mayol et al., 2012*). Although *A. balearica* has been cited (*Molins et al., 2011*) as an example of evolutionary stasis (low levels of morphological variation paralleled with low sequence variation), this has never been demonstrated.

*Arenaria balearica* is naturally distributed in Tyrrhenian islands of Majorca, Corsica, and Sardinia, including the surrounding minor islands of Tavolara, La Maddalena, Caprera, and Asinara, and in two of the main Tuscan Islands, Montecristo and Capraia

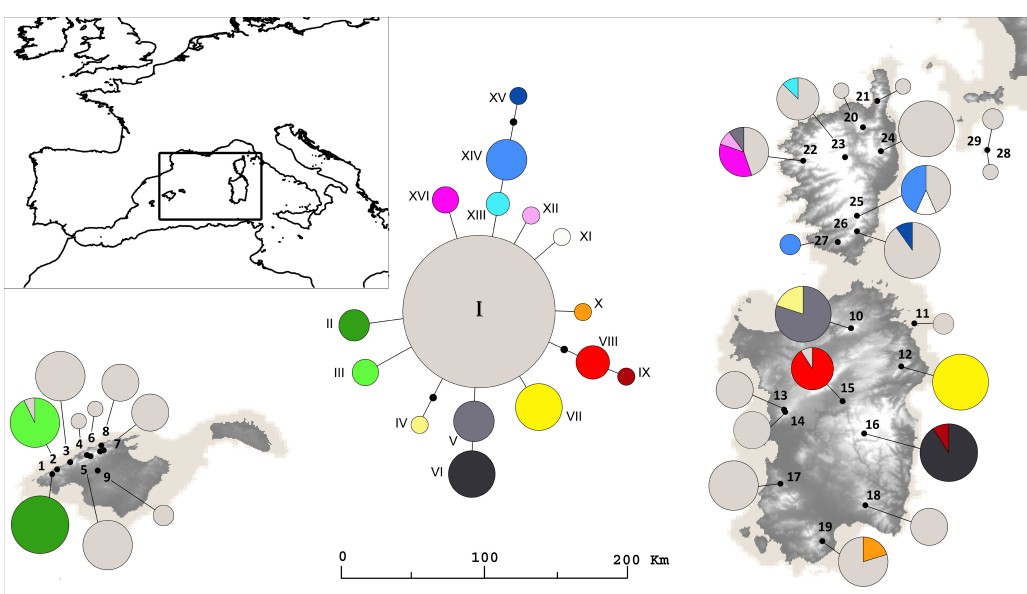

**Figure 1  Sampling localities and Haplotypes.** Sampling localities of *Arenaria balearica*, reconstruction of the coast line during the Last Glacial Maximum in the study area, spatial distribution of plastid DNA haplotypes and statistical parsimony network for 226 individuals. The small black circles represent missing intermediate haplotypes. Sectors within circles in the map indicate the presence of different haplotypes in different individuals of the same population.

(*Diana Corrias, 1981*). Most of the populations known from Majorca, Corsica and Sardinia are placed on the Hercynian basement of the corresponding island (*Alvarez, Cocozza & Wezel, 1974*; *Rosenbaum, Lister & Duboz, 2002*). The species is an alien plant in some European countries, where it is used as an ornamental. Due to its distribution pattern and to the fact that the plant usually inhabits plant communities having a notable relict character (*Bolòs & Molinier, 1958*), *A. balearica* has been traditionally considered to be a Mediterranean paleoendemic in the broad sense of the term (*Favarger & Contandriopoulos, 1961*), and a disjunct endemism by *Thompson (2005)*. The plant produces small seeds (0.5–0.6 mm) and although it lacks any evident adaptation to long-distance dispersal (LDD), such events due to stochastic mechanisms, even human mediated (*López González, 1990*), cannot be *a priori* ruled out to explain its current distribution pattern.

Previous studies on phylogeopraphic patterns of Mediterranean disjunct endemic species have focused on examples from the Eastern Mediterranean region (e.g., *Affre & Thompson, 1997*; *Widén, 2002*; *Bittkau & Comes, 2005*; *Edh, Widén & Ceplitis, 2007*), as well as from the Western Mediterranean region, including species distributed in Majorca and Menorca (e.g., *Sales et al., 2001*; *Molins, Mayol & Rosselló, 2009*) and Corsica and Sardinia (e.g., (*Falchi et al., 2009*). *Molins et al. (2011)* have studied *T. herba-barona*, a disjunct endemic that shows a distribution similar to that of *A. balearica* except for the facts that the former is not as widespread neither in Majorca (only one population) nor in Sardinia as *A balearica* and that it is absent from the islets of the Tuscan Archipelago.

Using both sequencing of plastid DNA regions and amplified fragment length polymorphism (AFLP) fingerprinting, this study aims to reconstruct the phylogeographic

patterns and differentiation of intraspecific lineages within the disjunct endemic plant *A. balearica*. More specifically our objectives are: (1) to test to which extent the observed distribution of *A. balearica* is concordant with the geological history of the continental fragment islands from the Western Mediterranean region; (2) to give a satisfactory answer to the question on how the colonization of the different islands and islets took place; and (3) to evaluate whether the low morphological variation observed among populations of *A. balearica* located in different islands is in correspondence with overall low levels of genetic diversity.

## MATERIALS AND METHODS

### Reconstruction of the coastline during the Last Glacial Maximum in the study area

During the Last Glacial Maximum (LGM), ice sheets covered large areas in northern latitudes, and global temperatures were significantly lower than today (*Yokohama et al., 2000*). At the LGM, the Earth's ocean levels were at their lowest point and extensive reaches of dry land were exposed along the continents' coasts. Several analyses have substantially narrowed the uncertainties regarding total changes in ice sheets and sea level and their proxies, suggesting a net decrease in the eustatic sea level at the LGM ranging from 120 to 135 m a.s.l. (*Church et al., 2001*; *Clark & Mix, 2002*). The reconstruction of coastlines at 21 Ka (kiloyears before present) for the study area presented here (Fig. 1) is derived from these references.

To map the past and current shorelines in detail, the present-day topographic and bathymetric data covering the area were taken from the ETOPO1, which is a 1 arc-minute global relief model of the Earth's surface that integrates land topography and ocean bathymetry. This model was built from numerous global and regional data sets, and is available in "Bedrock" (base of the ice sheets) versions (*NOAA, 2009*). Estimates of exposed land area at LGM with respect to the present-day are the result of the values of the Digital Elevation Model being raised by 120 m.

### Study species

*Arenaria balearica* is an herbaceous perennial delicate plant whose filiform, branched stems and small leaves form low, compact ever-green moss-like dense mats, preferentially on cool, moist soils in shaded rocky places (comophyte), although it can be secondarily found also on shady moist slopes, between 0 and 1,800 m a.s.l. (*Diana Corrias, 1981*; *López González, 1990*). Although there are no available data on the reproductive biology of the species, its slender, short, upright stems that bear white, actinomorphic flowers suggest that it is probably partly wind, and partly insect pollinated. Its chromosome number is $2n = 18$ (*Diana Corrias, 1981*; *López González, 1990*). Generation times are not known for the species. The available phylogenetic data based on the analysis of DNA sequences (*Fior & Karis, 2007*) indicate that this species is closely related to *Arenaria bertolonii* Fiori, which is distributed primarily in mainland Italy (*Iamonico, 2013*) and Sardinia (*Conti et al., 2005*). The most recent phylogeny published for the genus *Arenaria* L. (*Sadeghian et al., 2015*) concluded that. *A. balearica* should be excluded from *A.*sect. *Rotundifoliae* McNeill,
where the species was traditionally included. Unfortunately, these authors did not include *A. bertolonii* in the phylogeny and recovered *A. balearica* in a largely unresolved position (very low levels of statistical support).

## Sampling strategy, outgroup selection and monophyly test

Leaf material from a total of 250 plants from 29 sampling sites including the islands of Majorca (9), Corsica (8), Sardinia (9), Tavolara (1), and Montecristo (2), representing the entire distribution range of *A. balearica*, was collected and dried in silica gel (Table 1 and Fig. 1). Each sampling site was geo-referenced with a GPS GARMIN GPSMAP 60, and voucher specimens were deposited at the herbaria of the University of Salamanca (SALA), of the University of Granada (GDA) in Spain and/or of the University of Cagliari (CAG) in Sardinia, Italy.

The intent was to include a minimum of 10–12 plants per population in the analysis, but sometimes the population sizes were small and it was not possible to collect such a quantity of well separated (>5–10 m) individuals. Also, further problems were encountered in some cases in the DNA extraction and amplification processes (the leaves are only 2–4 mm and it was many times difficult to get an adequate quantity of DNA). In this situation, a variable number of 1–16 plants per sampling site were finally used (Table 1).

Three additional samples from *A. bertolonii* were selected to be used as outgroup in the plastid DNA haplotype analyses. Given the uncertain phylogenetic position of *A. balearica* within the genus according to the most recent data (*Sadeghian et al., 2015*), the selection of this outgroup was based on the results by *Fior & Karis (2007)*. Furthermore, the monophyly of the study group was assessed in a parallel study (J Bobo-Pinilla, J Peñas de Giles & MM Martínez-Ortega, 2013, unpublished data) through the phylogenetic analysis of nucleotide sequences of the nuclear ribosomal internal transcribed spacer (ITS) using 28 samples belonging to *A. balearica* and several other samples from the related species *A. funiculata* (Pau) Fior & P.O. Karis, *A. tejedensis* (Willk.) Fior & P.O. Karis and *A. suffruticosa* Fior & P.O. Karis. These data further support the sister group relationship between *A. balearica* and *A. bertolonii* already proposed by *Fior & Karis (2007)*.

## DNA isolation, AFLP amplification, and data analysis

Total genomic DNA was isolated from crushed dried leaf material (ca. 25 mg) following the 2× CTAB (cetyl trimethyl ammonium bromide) protocol (*Doyle & Doyle, 1987*) with minor modifications. The quality of the extracted DNA was checked in 1% TAE-agarose gel. A negative control sample was consistently included to test for contamination, and five randomly chosen samples were replicated to test for reproducibility.

Given the very small leaf size of *A. balearica*, it was not always possible to extract enough DNA to provide clear and reliable AFLP profiles. Therefore, five populations among the 29 initially sampled had to be excluded from the AFLP analysis (Table 1). AFLP profiles were finally drawn for 213 individuals following established protocols (*Vos et al., 1995*). An initial screening of selective primers was performed using 26 primer combinations. The four finally selected primer combinations (fluorescent dye in brackets), (6-FAM)*Eco*RI-ACT/*Mse*I-CAT, (6-FAM)*Eco*RI-AGA/*Mse*I-CTG, (VIC)*Eco*RI-AAG/*Mse*I-CAT, (VIC)*Eco*RI-AGG/*Mse*I-CC, were used for the selective polymerase chain

**Table 1 Sampling localities and genetic data.** Population names and sampling localities, AFLP descriptors and plastid DNA haplotypes for the studied populations of Arenaria balearica.

| Sampling locality | DIYABC assignation | Elevation (m a.s.l.) | Long./Lat. | $N_{AFLP}$ | Nei's GD | DW | $H_{cp}$ |
|---|---|---|---|---|---|---|---|
| 1: SP; Majorca, Estellencs, Puig de Galatzó | MAJ | 962 | 2.48°/39.63° | 11 | 0.096 | 5.872 | I (1); II (11) |
| 2: SP; Majorca, Banyalbufar, Mola de Planicia | MAJ | 726 | 2.52°/39.67° | 10 | 0.098 | 4.491 | II (9) |
| 3: SP; Majorca, Valldemossa, Puig des Teix | MAJ | 906 | 2.63°/39.73° | 10 | 0.119 | 6.775 | I (9) |
| 4: SP; Majorca, Escorca, Puig Major | MAJ | 847 | 2.77°/39.79° | – | – | – | I (2) |
| 5: SP; Majorca, Escorca, Tossals | MAJ | 972 | 2.80°/39.78° | 10 | 0.110 | 5.625 | I (9) |
| 6: SP; Majorca, Escorca, Clot d'Albarca | MAJ | 468 | 2.88°/39.82° | – | – | – | I (1) |
| 7: SP; Majorca, Escorca, Puig Tomir | MAJ | 882 | 2.91°/39.83° | 10 | 0.189 | 14.83 | I (7) |
| 8: SP; Majorca, Escorca, Puig Caragoler | MAJ | 753 | 2.89°/39.87° | 8 | 0.095 | 7.083 | I (7) |
| 9: SP; Majorca, Escorca, Puig d'en Galileu | MAJ | 879 | 2.85°/39.81° | 9 | 0.119 | 5.653 | I (4) |
| 10: IT; Sardinia, Tempio Pausania, Madonna del Limbara - Monte Limbara | NSA | 1,230 | 9.16°/40.85° | 10 | 0.167 | 8.518 | IV (2); V (8) |
| 11: IT; Sardinia, Olbia, Tavolara | - | 470 | 9.69°/40.89° | 5 | 0.179 | 13.208 | I (5) |
| 12: IT; Sardinia, Lula, Punta Turuddò - Monte Albo | NSA | 1,094 | 9.58°/40.53° | 9 | 0.161 | 8.894 | VII (10) |
| 13: IT; Sardinia, Cuglieri, La Madonnina | SSA | 802 | 8.60°/40.17° | 8 | 0.135 | 7.506 | I (6) |
| 14: IT; Sardinia, Santu Lussurgiu, Zorzia - Monte Urtigu | SSA | 978 | 8.61°/40.15° | 9 | 0.151 | 6.808 | I (7) |
| 15: IT; Sardinia, Oliena, Monte Corrasi | NSA | 980 | 9.09°/40.24° | 9 | 0.136 | 6.936 | I (1); VIII(7) |
| 16: IT; Sardinia, Desulo, Taccu di Girgini | NSA | 120 | 9.27°/39.97° | 16 | 0.103 | 5.664 | VI (10); IX (1) |
| 17: IT; Sardinia, Guspini, Montevecchio | SSA | 276 | 8.57°/39.55° | 8 | 0.179 | 9.914 | I (9) |
| 18: IT; Sardinia, Burcei, Rio Niu Crobu - Monte Serpeddi | NSA | 856 | 9.28°/39.37° | 10 | 0.129 | 6.700 | I (7) |
| 19: IT; Sardinia, Villa S. Pietro, Rio Is Canargius - Monte Nieddu | SSA | 183 | 8.92°/39.07° | 10 | 0.123 | 6.744 | I (7); X (2) |
| 20: FR; Corsica, Cap Corse, Commune d'Olmeta | COR | 800 | 9.69°/42.75° | – | – | – | I (1) |
| 21: FR; Corsica, Massif de Monte Astu, 1.25 km NW Lento | COR | 1025 | 9.26°/42.53° | – | – | – | I (1) |

**Table 1** (*continued*)

| Sampling locality | DIYABC assignation | Elevation (m a.s.l.) | Long./Lat. | N_{AFLP} | Nei's GD | DW | H_{cp} |
|---|---|---|---|---|---|---|---|
| 22: FR; Corsica, Gorges de Spelunca, Le Sentier de la Spilonca | COR | 233 | 8.76°/42.25° | 10 | 0.187 | 10.881 | I (4); V (1); XII (1); XVI (3) |
| 23: FR; Corsica, Valle de la Restonica | COR | 492 | 9.11°/42.28° | 8 | 0.182 | 13.227 | I (7); XIII (1) |
| 24: FR; Corsica, Valle de'Alesani, Quercetto | COR | 677 | 9.41°/42.33° | 10 | 0.163 | 9.794 | I (10) |
| 25: FR; Corsica, Col de Bavella | COR | 1,317 | 9.21°/41.79° | 9 | 0.169 | 8.681 | I (4); XI (1); XIV (4) |
| 26: FR; Corsica, La Cascade de Piscia di Ghjadu | COR | 209 | 9.21°/41.66° | 10 | 0.194 | 13.182 | I (9); XV (1) |
| 27: FR; Corsica, Gianuccio | COR | 537 | 9.05°/41.57° | 4 | 0.200 | 12.906 | XIV (3) |
| 28: FR; Tuscan Archipelago, Montecristo, Collo a fundo | – | 460 | 10.31°/42.32° | – | – | – | I (1) |
| 29: FR; Tuscan Archipelago , Montecristo, Grotta del Santo | - | 251 | 10.30°/42.34° | – | – | – | I (2) |

SP, Spain; IT, Italy; FR, France; NAFLP, number of individuals investigated with AFLP; Nei's GD, *Nei*'s *(1987)* gene diversity; DW, frequency down-weighted marker values; HCP, plastid DNA (cpDNA) haplotypes derived from concatenated sequences, the number of individuals per haplotype per population is given in parentheses.

reaction. These combinations were selected because they generated a relatively high number of clearly reproducible bands. A relatively high number of alleles per individual is desirable, given that AFLP are dominant markers (*Lowe, Harris & Ashton, 2004*). Samples (3 μl) of the fluorescence-labelled selective amplification products were combined and separated on a capillary electrophoresis sequencer (ABI 3730 DNA Analyser; Applied Biosystems, Foster City, CA, USA), with GenScan ROX (Applied Biosystems) as an internal size standard.

Raw AFLP data with amplified fragments from 150 to 500 base pairs (bp) were scored and exported as a presence/absence matrix using the software GeneMapper 4.0 (Applied Biosystems). As an initial approach to the global genetic relationships among the individuals analysed and possible structure of the data, a Neighbour-Joining (NJ) analysis including 1,000 bootstrap pseudoreplicates based on a matrix of Nei-Li (*Nei & Li, 1979*) distances was conducted with the software Paup 4.0b10 (*Swofford, 2003*). An unrooted NeighbourNet was also produced using the program SplitsTree 4.12.3. *Huson & Bryant (2006)* and based on Dice's coefficient, which is suitable for multilocus dominant genetic data (*Dice, 1945*; *Lowe, Harris & Ashton, 2004*). Additionally, a Principal Coordinate Analysis (PCoA) based on a matrix of Dice's coefficient among individuals was performed with NTSYS-pc 2.02 (*Rohlf, 2009*).

Population genetic structure was additionally investigated using a Bayesian clustering method implemented in STRUCTURE v. 2.3.4 (*Pritchard, Stephens & Donnelly, 2000*) following the approach described by *Falush, Stephens & Pritchard (2007)* for dominant markers. This method uses a Markov chain Monte Carlo simulation approach to group samples into an optimal number of *K* genetic clusters and does not assume an *a priori* assignment of individuals to populations, nor to clusters. Analyses were based on an ancestral admixture model with correlated allele frequencies among populations. The

proportion of membership of each individual and population to the $K$ clusters was calculated by performing 20 runs for each $K$ value between 2 and 9 with a run length of the Markov chain Monte Carlo of $1 \times 10^6$ iterations after a burn-in period of $1 \times 10^6$ iterations, with $\lambda$ adjusted at 0.4523. The optimal number of $K$ clusters was estimated using the *ad hoc* parameter ($\Delta K$ statistic) of *Evanno, Regnatus & Goudet (2005)*, as implemented in the online application of Structure Harvester software (v0.63; *Earl & Vonholdt, 2012*).

Although aware that AFLP-based estimates of the level of genetic variation could be biased in this case by low sampling sizes and relative differences in sampling effort, *Nei*'s (*1987*) gene diversity index was calculated for each population (or sampling site) using the R package AFLPDAT (*Ehrich, 2006*). This package was also used to calculate the frequency down-weighted marker values per population or sampling site (DW; *Schönswetter & Tribsch, 2005*), which is an estimation of the genetic rarity of a population.

To test the comparative historical effects of the main biogeographical barriers, a hierarchical analysis of molecular variance (AMOVA) was performed with the software ARLEQUIN 3.5.1.2 (*Excoffier & Lischer, 2010*). For this, genetic variation was distributed into portions assignable to differences among predefined geographical groups ($F_{CT}$), among populations within these groups ($F_{SC}$), and among populations across the entire study area ($F_{ST}$) (*Turner et al., 2000*; *Ortiz et al., 2009*). Additionally, four alternative groupings were tested using AMOVA analysis: the first two tested the groups derived from PCoA and NJ analyses, respectively, while the third and fourth ones tested two additional geographical groupings (i.e., (Majorca) (Corsica) (Sardinia + Tavolara) and (Majorca) (Corsica + Sardinia + Tavolara), respectively).

## Plastid DNA sequencing and data analysis

Three regions of the plastid DNA were sequenced and haplotype variation was explored to complement the information given by the mainly nuclear AFLPs. The plastid regions *trn*L$^{UAA}$-*trn*F$^{GAA}$ (*Taberlet et al., 1991*), *psbA-3′ trnK-matK* and *rpS16* (*Shaw et al., 2005*) showed the highest variability among seven surveyed regions (*trnQ(UUG)-rps16x1, trnL-rpl32F, atpI-atpH, Shaw et al., 2007*; *rpoB-trnC, trnH-psbA, Shaw et al., 2005*) and were used to analyse a total of 226 plants from 29 populations (Table 1) of *A. balearica*. PCR conditions and primers for DNA amplification are detailed in Table 2. PCR products were visualized on 1% agarose gel and purified using PCR Clean-Up with ExoSAP-IT Kit (AFFIMETRIX, Santa Clara, CA, USA) following the manufacturer's instructions. The cleaned amplification products were analysed with a 3730 DNA Genetic Analyser capillary sequencer (Applied Biosystems). All sequences can be found in the Supplemental Information (Data S2 and S3).

Congruence in the phylogenetic signal of the different plastid DNA regions was tested with the partition homogeneity test (ILD; *Farris et al., 1995a*; *Farris et al., 1995b*). ILD significance values were calculated in TNT v.1.1 (*Goloboff, Farris & Nixon, 2003*) with the INCTST script—kindly provided by the authors of the program—with 1,000 replicates. The plastid DNA sequences were assembled and edited using GENEIOUS PRO™ 5.4 (*Drummond et al., 2012*) and aligned with CLUSTALW2 2.0.11 (*Larkin et al., 2007*); further adjustments and optimisations were made by visual inspection. Sequences from the three regions were

**Table 2** **PCR values.** PCR primers and conditions used to obtain plastid DNA sequence data for *A. balearica*; number of substitutions (*S*) and number of indels (*I*).

| cpDNA region | Forward primer | Reverse primer | Denaturation Temperature/Time | Annealing Temperature/Time | Extension Temperature/Time | Cycles | *S* | *I* |
|---|---|---|---|---|---|---|---|---|
| trnL$^{UAA}$-trnF$^{GAA}$[a] | tabC | tabF | 95 °C/30″ | 57 °C/30″ | 72 °C/1′30″ | 35 | 3 | 11 |
| psbA-′ trnK-matK [b] | matK8F | psbA5′R | 95 °C/30″ | 52 °C/30″ | 72 °C/1′30″ | 35 | 3 | 3 |
| rpS16[b] | rpS16F | rpS16R | 95 °C/30″ | 55 °C/30″ | 72 °C/1″30″ | 35 | 5 | 8 |

**Notes.**
[a] *Taberlet et al. (1991)*.
[b] *Shaw et al. (2005)*.

concatenated based on the assumption that the plastid forms a single linkage group into a single matrix to be analysed, considering also that the ILD test did not report significant incongruities among DNA regions. Gaps (insertions/deletions) were coded as single-step mutations and treated as a fifth character state. Mononucleotide repeats of different sizes were excluded given that they seem to be prone to homoplasy at large geographic scales (*Ingvarsson, Ribstein & Taylor, 2003*).

The completeness of haplotype sampling across the range of *A. balearica* was estimated using the Stirling probability distribution. It provides a way to evaluate the assumption that all haplotypes have been sampled (*Dixon, 2006*).

As an approach to infer the genealogical relationships among haplotypes, an unrooted haplotype network was constructed using the statistical parsimony algorithm (*Templeton, Crandall & Sing, 1992*) as implemented in TCS 1.21 (*Clement, Posada & Crandall, 2000*).

Six competing phylogeographic hypotheses were compared using a coalescent based approximate Bayesian computation method (ABD approach), as implemented in DIYABC v2.1 software (*Cornuet et al., 2014*). DIYABC allows testing the posterior probabilities of alternative scenarios involving complex population histories (i.e., any combination of population divergences and multifurcations, admixture events, population size changes, bottlenecks, etc., even with population samples potentially collected at different times and/or with unsampled populations, *Cornuet et al., 2014*). The logistic regression procedure (*Fagundes et al., 2007*) gives an estimate of the occurrence of each scenario among simulated data sets that are closest to the observed data. In our case, four different metapopulations (i.e., Majorca, Corsica, NE Sardinia and SW Sardinia, correspondingly MAJ, COR, NSA and SSA in Table 1) were considered. Due to low sample sizes and considering that only the most widely represented haplotype was present, populations 11, 28 and 29 were excluded from this analysis in order to avoid increasing exponentially computation times . The distinction between NE Sardinia and SW Sardinia (Table 1) was made considering relevant geological aspects, particularly the fact that the populations of *A. balearica* present in the island are located exclusively on two different geological units both located on the ancient Hercynian basement of the island and mainly separated by Oligocene and Miocene rift basins and Plio-Pleistocene basalts (*Rosenbaum, Lister & Duboz, 2002*). After some initial analysis and taking into account the haplotype network, the geographical distribution of the species and these geological aspects, six competing

phylogeographic scenarios were designed. A list of all parameters and prior distributions used to model scenarios is summarized in Table 3. Prior distributions of the parameters were chosen as a first approach with a large interval due to the lack of ancestral information. Parameters were subsequently corrected according to values obtained after first tests. Population sizes were set equally in all cases; divergence times were taken unrestricted to allow the program to set the most likeable value. Uniform Mutation rate was set to ($10^{-9}$–$10^{-7}$). One million data sets were simulated for each scenario (Cornuet et al., 2008; Cornuet, Ravigné & Estoup, 2010). The posterior probabilities of each one were calculated by performing a polychotomous weighted logistic regression on the 1% of simulated data sets closest to the observed data set (Cornuet et al., 2008; Cornuet, Ravigné & Estoup, 2010). The posterior distributions of parameters were evaluated under the best scenario using a local linear regression on the 1% closest simulated data sets with a logit transformation (Table 3). Bias and precision for the parameters estimations were also calculated. Divergence time between groups must be taken carefully, due to the lack of information about generation times for the species. Confidence in scenario choice has been tested by evaluating Type I and Type II error rates (Cornuet, Ravigné & Estoup, 2010).

## RESULTS

### Population structure based on AFLP

The four primer combinations applied to 213 plants representative of the variation of the species *A. balearica* produced a total of 792 reproducible fragments.

Both the NJ and NeighbourNet diagrams conducted on all individuals revealed a relatively weak overall structure of the genetic variation into two main groups: one comprised the samples collected in Majorca ("group 1", represented in green in Fig. 2A; populations 1–3, 5, 7–9; with not significant bootstrap support, BS < 75%) and a second poorly supported group (BS < 75%), which clustered together individuals from the remaining populations included in this study. Within the second group, three further subgroups were found: first, "group 2," which included samples collected mostly in C and S Sardinia (populations 14, 15, 18 and 19); second, "group 3," which grouped populations 10–13, plus 17 from W and NE Sardinia and Tavolara, together with populations 23–27 mostly from S Corsica; and third, "group 4," which included all the individuals from population 16 in C Sardinia. None registered significant BS values (BS < 75%).

Apparently a higher level of overall genetic structure was revealed by the PCoA (Fig. 2B); in this case, the first two axes accounted for 55.31% and 5.33%, respectively, of the total variance, although no evident geographic structure was found. Two groups were roughly distinguished in the PCoA: the first one grouped populations 1–3, 5, 7–9 from Majorca with 10, 12, 15, 16, and 19 from Sardinia, while the second contained populations 11, 13, 14, 17, and 18 from Sardinia and Tavolara, with 22–27 from Corsica. This analysis indicated differentiation to a certain degree of the populations from Majorca and Corsica, but not of those from Sardinia or Tavolara. The genetic structure revealed by NJ and PCoA did not coincide except for the fact that the populations from Majorca were slightly differentiated from the Corso-Sardinian ones.

**Table 3  Parameters used in DIYABC analyses.**

| Parameter | Scenario | Parameter code | Prior Distribution | | | Estimated Parameters | |
|---|---|---|---|---|---|---|---|
| | | | Type | Initial Interval | Final Interval | Mean | Median |
| Population effective sizes of the MAJ group | All | Nmaj | Uniform | {10–100.000} | {10–6.000} | 4.500 | 4.490 |
| Population effective sizes of the COR group | All | Ncor | Uniform | {10–100.000} | {10–30.000} | 24.700 | 26.100 |
| Population effective sizes of the NSA group | All | Nnsa | Uniform | {10–100.000} | {10–5.000} | 1.790 | 1.940 |
| Population effective sizes of the SSA group | All | Nssa | Uniform | {10–100.000} | {10–18.000} | 16.000 | 16.600 |
| Founder event for MAJ group | | NFmaj | Uniform | {10–500} | {10–500} | | |
| Founder event for COR group | | NFcor | Uniform | {10–500} | {10–500} | | |
| Founder event for NSA group | | NFnsa | Uniform | {10–500} | {10–500} | | |
| Founder event for SSA group | | NFssa | Uniform | {10–500} | {10–500} | | |
| Divergence time corresponding to ancestral area fragmentation | 1 | T1 | Uniform | {10–1.000.000} | {10–10.000} | 4.640 | 4.730 |
| Divergence time between NSA and SSA | 2 & 5 | T2 & T9 | Uniform | {10–1.000.000} | {10–20.000} | | |
| Divergence time corresponding to diferenciation into three main islands | 2 | T3 | Uniform | {10–1.000.000} | {10–20.000} | | |
| Divergence time between COR and NSA | 3 | T4 | Uniform | {10–1.000.000} | {10–30.000} | | |
| Divergence time between SSA and MAJ | 3 | T5 | Uniform | {10–1.000.000} | {10–15.000} | | |
| Divergence time betwen [SSA+MAJ] and [COR+NSA] | 3 | T6 | Uniform | {10–1.000.000} | {10–40.000} | | |
| Divergence time between COR and MAJ | 4 | T7 | Uniform | {10–1.000.000} | {10–10.000} | | |
| Divergence time among [COR+MAJ], SSA and NSA | 4 | T8 | Uniform | {10–1.000.000} | {10–20.000} | | |
| Divergence time between COR and Sardinia | 5 | T10 | Uniform | {10–1.000.000} | {10–10.000} | | |
| Divergence time betwen MAJ and [NSA, SSA and COR] | 5 | T11 | Uniform | {10–1.000.000} | {10–20.000} | | |
| Divergence time among groups in Corsica and Sardinia | 6 | T12 | Uniform | {10–1.000.000} | {10–15.000} | | |
| Divergence time for initial isolation of MAJ | 6 | T13 | Uniform | {10–1.000.000} | {10–20.000} | | |
| Mean mutation rate | All | Mμ | Uniform | $\{10^{-9}–10^{-7}\}$ | $\{10^{-9}–10^{-7}\}$ | 6,48E − 08 | 6,44E − 08 |

Nei's gene diversity index (Table 1) ranged from 0.09 (populations 8, 1, and 2, all from Majorca) to 0.20 (population 27 from Corsica, although this result may be biased due to the small sampling size) and DW varied between 4.49 in population 2 and 14.83 in population 7, both from Majorca. Overall, the genetically most distinctive and diverse populations were found in Corsica, while the populations from Majorca displayed generally low diversity and singularity values.

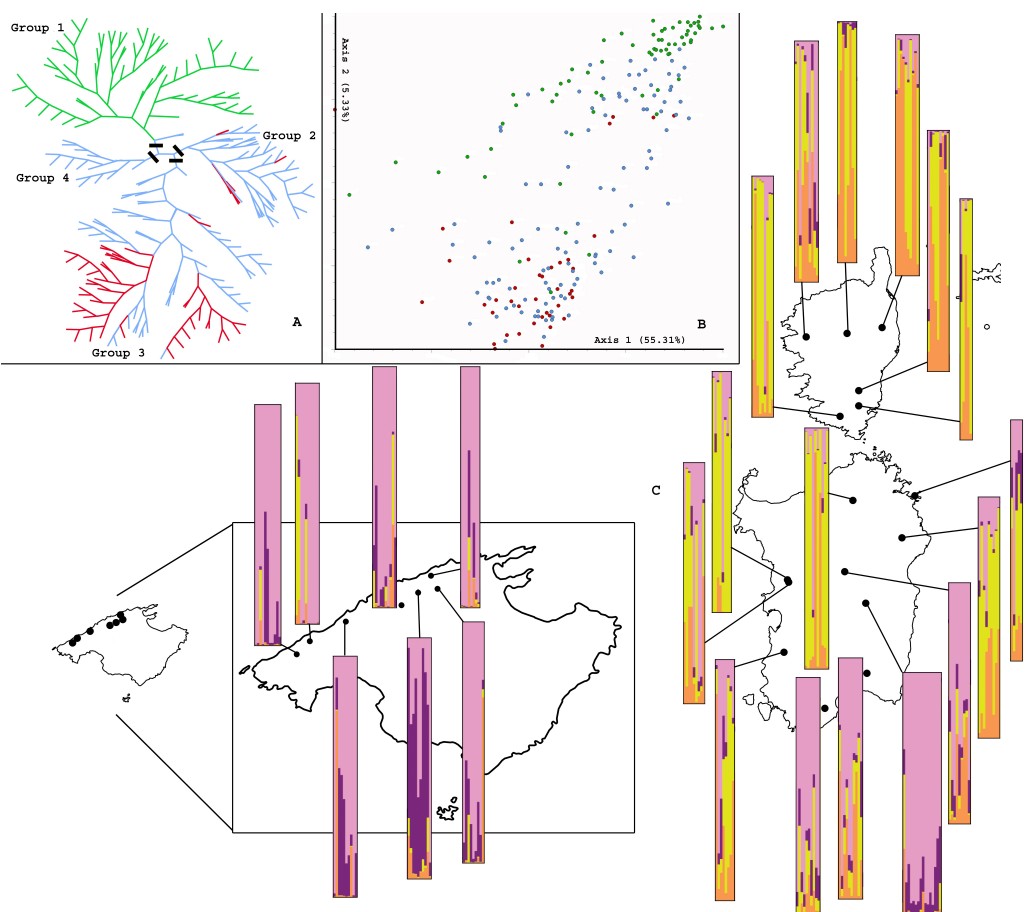

**Figure 2   AFLP results.** Genetic population structure based on AFLP analysis of 213 individuals of *Arenaria balearica*: (A) Unrooted neighbour-joining analysis; colours correspond to islands: branches in green lead to individuals from Majorca, in red to individuals from Corsica, in blue to individuals from Sardinia; the four groups commented in the text are indicated with a black line. (B) Ordination of AFLP data according to a Principal Coordinates Analysis; colours corresponding to islands as in (A). (C) Admixture analysis conducted with the software Structure: the graphs next to each population projected in the map indicate the proportional assignment of individuals to the genetic clusters A (pink), B (purple), C (yellow) and D (orange).

Bayesian clustering conducted using STRUCTURE estimated $K = 4$ as the most likely number of genetic clusters in *A. balearica*, with a maximum modal value of $\Delta K = 12.414075$ (Fig. 3). This clustering (Fig. 2) showed that all four of these groups were represented in the three main islands and also in Tavolara. In summary, Cluster A (pink) was dominant in the populations from Majorca and S Sardinia (particularly in population 16), was well represented in Tavolara, but its representation was poor in the remaining populations, particularly in populations 23, 25, and 26 from Corsica; Cluster B (purple) was also well represented—but consistently in a lower proportion than Cluster A—in Majorca (especially in population 5), southern Sardinia (particularly in population 16) and Tavolara, but it was present in a very low proportion in the remaining populations included in this study; Cluster C (yellow) was very well represented in all populations from Corsica, northern

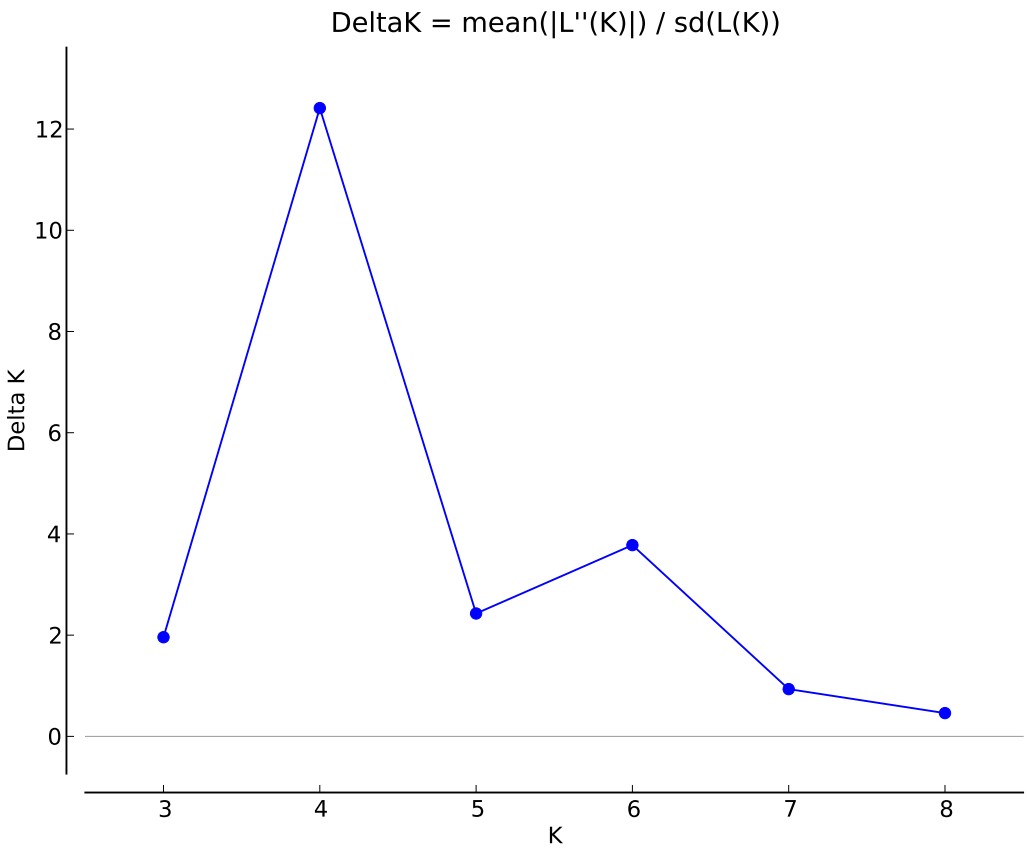

$$\text{DeltaK} = \text{mean}(|L''(K)|) / \text{sd}(L(K))$$

**Figure 3** **Delta *K* values from the method by** *Evanno, Regnatus & Goudet (2005)*.

Sardinia, and Tavolara, but was almost absent from Majorca (completely absent from population 3); and Cluster D (orange) was best represented in Corsica, was present also in Tavolara and Sardinia (in an almost insignificant proportion in population 16), and had also a low representation in Majorca.

The hierarchical AMOVA (Table 4) showed that the genetic structure in four groups detected by NJ (i.e., (populations 1, 2, 3, 5, 7, 8, 9) (populations 14, 15, 18, 19, 22) (populations 10–13, 17, 23–27) (population 16)) accounted for a comparatively higher amount of the total genetic variance (10.71%), among these groups. This amount was similar, although slightly lower, than that accounted for among populations within groups (11.41%). In the AMOVA analyses that evaluated other groupings the levels of genetic divergence were remarkably low among all groups considered and most of the variation was consistently found among populations within groups instead of among pre-established groups.

### Plastid DNA variation in *Arenaria balearica* and geographical distribution of haplotypes

The length of the three plastid DNA regions for 226 individuals ranged between 846 and 704 bp, and resulted in an alignment of 2291 bp, 17 polymorphisms (12 substitutions/five indels) were detected across the whole dataset, five (four substitutions/1 indels), eight (four

**Table 4 AMOVA analysis.** Comparison of analyses of molecular variance (AMOVA) based on AFLP data. Groupings of populations are shown in brackets (see text).

| Source of variation | *d.f.* | Sum of squares | Variance components | Variance % | *F*-values | 95% confidence interval |
|---|---|---|---|---|---|---|
| *Arenaria balearica* | | | | | | |
| Populations | 22 | 9274.91 | 31.72 | 19.80 | $F_{ST}$: 0.198 | |
| Individuals | 190 | 24415.52 | 128.50 | 80.20 | | |
| *Grouping 1 (PCoA derived):* [1,2,3,5,7,8,9,10,12,15,16,19] [11,13,14,17,18,22–27] | | | | | | |
| Groups | 1 | 1670.77 | 12.49 | 7.51 | $F_{CT}$: 0.075 | 0.064–0.083 |
| Populations | 21 | 7604.15 | 25.32 | 15.22 | $F_{SC}$: 0.165 | |
| Individuals | 190 | 24,415.52 | 128-50 | 77.27 | $F_{ST}$: 0.227 | |
| *Grouping 2 (NJ derived):* [1,2,3,5,7,8,9] [14,15,18,19,22] [10-13,17,23–27] [16] | | | | | | |
| Groups | 3 | 3652.31 | 17.66 | 10.71 | $F_{CT}$: 0.107 | 0.096–0.117 |
| Populations | 19 | 5622.61 | 18.82 | 11.41 | $F_{SC}$: 0.128 | |
| Individuals | 190 | 24,415.52 | 128.50 | 77.89 | $F_{ST}$: 0.221 | |
| *Grouping 3 (main islands, Sardinia includes Tavolara):* [1,2,3,5,7,8,9] [10–19] [22–27] | | | | | | |
| Groups | 2 | 2805.75 | 15.57 | 9.42 | $F_{CT}$: 0.094 | 0.084–0.104 |
| Populations | 20 | 6469.17 | 21.19 | 12.82 | $F_{SC}$: 0.141 | |
| Individuals | 190 | 24,415.52 | 128.50 | 77.76 | $F_{ST}$: 0.222 | |
| *Grouping 4: 2 groups, Majorca vs. Corsica+Sardinia+ Tavolara* [1,2,3,5,7,8,9] [10–27] | | | | | | |
| Groups | 1 | 1897.31 | 16.55 | 9.78 | $F_{CT}$: 0.098 | 0.081–0.110 |
| Populations | 21 | 7377.61 | 24.19 | 14.29 | $F_{SC}$: 0.158 | |
| Individuals | 190 | 24,415.52 | 128.50 | 75.93 | $F_{ST}$: 0.240 | |

substitutions/four indels) and four substitutions were detected for the $trnL^{UAA}$-$trnF^{GAA}$, *psbA-3′ trnK-matK* and *rpS16*, respectively. All mutations together defined a total of 16 haplotypes (Table 1). The results of the ILD test did not reveal significant inconsistencies among the plastid-DNA regions studied. The completeness of haplotype sampling estimated using *Dixon*'s (*2006*) method was 0.97 (the most likely value of haplotypes = 16), suggesting that all haplotypes present in the species had been sampled.

The statistical parsimony algorithm implemented in TCS inferred a 95% parsimony network with a maximum limit of four steps and star-like topology (Fig. 1). As inferred from the networking analysis, *A. balearica* showed a single major haplotype (present in 24 from the 29 populations studied), probably ancestral (haplotype I), which occurred in all islands (including Tavolara and Montecristo). In addition, there were 15 haplotypes, nine haplotypes (II, III, V, VII, X, XI, XII, XIII and XVI) separated one step from the ancestral one, haplotypes VI and XIV derived one step from haplotypes V and XIII respectively and haplotype XV derived two steps from XIV, two haplotypes derived two steps from haplotype I (IV and VIII) and IX derived one step from VIII. The most derived haplotypes were endemic to one individual island and usually were restricted to single populations (except for haplotype XIV, which was found in two populations from Corsica). Apart from haplotype I, only haplotype V was shared by populations located in different islands (Corsica and Sardinia). *Arenaria bertolonii* is separated 50 steps from the *A. balearica* central haplotype. The levels of haplotypic variation found in Corsica and Sardinia seems to be in accordance with the high levels of overall genetic diversity revealed by AFLP markers.

### DIYABC analysis

Scenario 1 (ancestral area fragmentation) was revealed as the most probable. The posterior probability of the logistic regression was 75%, while the alternative hypotheses (Fig. 4) received less than 7%. Type I and type II errors corresponding to Scenario 1 resulted to be 21% and 17% respectively. DIYABC software places the fragmentation of the four areas 4730 generations ago.

## DISCUSSION

### Phylogeography of the relict *Arenaria balearica*

Solid analysis in phylogeography should be based on the choice of appropriate study organisms and focal areas. Several requirements for reliable phylogeographic inference should be met, among them a sound phylogenetic framework and the absence of obvious adaptations for LDD from the organism side, and the availability of good historical climatic and geographic data from the focal-area side (*Salvo et al., 2010*). *Arenaria balearica* and the Western Mediterranean region satisfy these prerequisites. One of the most basic questions related with Mediterranean plant populations that still remains open is what part of their present genetic diversity is, as generally assumed, due to isolation in refugia during the Pleistocene glaciations and what part can be traced back to the Tertiary history of taxa (*Magri et al., 2007*; *Médail & Diadema, 2009*). Several authors (*Thompson, 2005*; *Donoghue, 2008*; *Ackerly, 2009*) have suggested that the filtering of elements from the ancient Tertiary geofloras that spread across the Northern Hemisphere during the Tertiary (*Wolfe, 1975*; *Wolfe, 1978*) played a crucial role in the assembly of the Mediterranean floristic diversity. Thus, traditionally, botanists have classified the floristic elements of the Mediterranean region into two main groups, depending on whether these were believed to have arisen before or after the development of Mediterranean-like climates (*Thompson, 2005*; *Salvo et al., 2010*). *Arenaria balearica* was traditionally considered a Tertiary relict palaeoendemic species (*Contandriopoulos, 1962*) and has been particularly mentioned as a "Hercynian palaeoendemic" (*Molins et al., 2011*). Unfortunately, considering that the plant is perennial and that there is no information available on generation times, although we have obtained here an estimated divergence time for T1 (Table 3; Fig. 4), our results are not conclusive regarding the question on the age and hypothetic ancient origin of the species.

Several hypotheses may explain the presence of *A. balearica* in Majorca, Corsica, and Sardinia, plus minor Tyrrhenian continental fragment islands. This striking distribution may suggest that it could be a non-monophyletic lineage, but the phylogenetic analysis of ITS (nrDNA) and plastid DNA sequences, which included samples from all the Tyrrhenian islands where the species is represented, indicated that the study group is clearly monophyletic (J Bobo-Pinilla, J Peñas de Giles & MM Martínez-Ortega, 2013, unpublished data). Additionally, both the careful review of herbarium materials prior to the sampling performed within this study, as well as the field observations, indicate very low morphological variation among populations (J Lorite, 2014, unpublished data).

Both plastid and nuclear markers show the lack of a phylogeographic break among populations from different islands. Low levels of genetic structure are repeatedly found

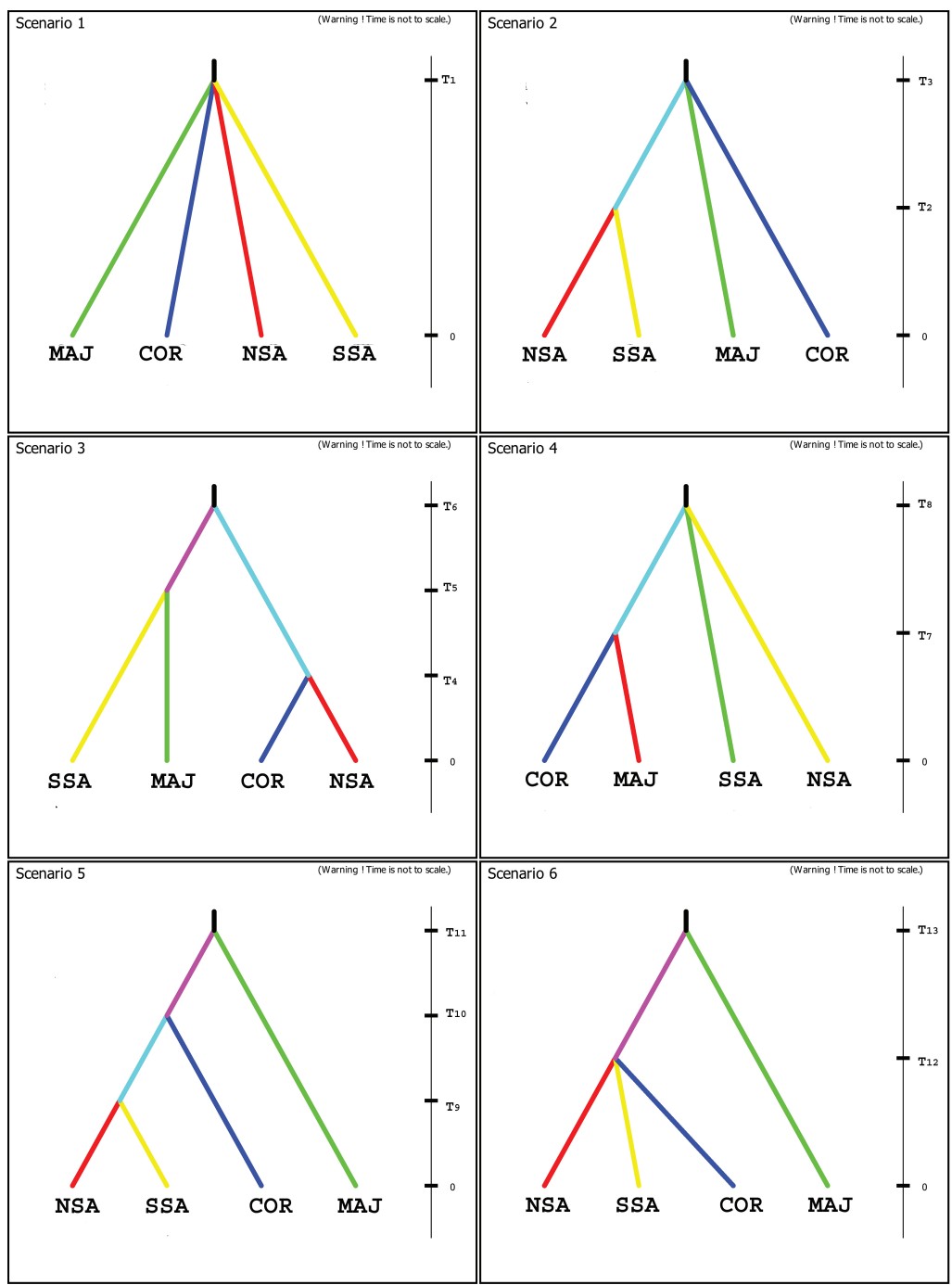

**Figure 4  Scenarios used in DIYABC.** Graphic representation of the 6 scenarios used in DIYABC.

by the data analyses derived from the anonymous, mostly nuclear, DNA fingerprints (i.e., AFLP data; NJ, NNet and PCoA analyses; Fig. 2) and by the plastid-DNA data. The AMOVA analyses also indicated moderate levels of divergence among populations of *A. balearica* considered as a unique group, which are even lower among the different groups tested with

AMOVA. These results contrast with the expectation of high population or geographical group divergence in species that occur in spatially isolated territories, particularly when the species shows limited dispersal abilities (in these situations gene flow tends to be low and, especially when population sizes are small, the effect of genetic drift is usually high). In the case of *A. balearica*, the moderate levels of divergence found may represent remnants of Messinian contacts among the Tyrrhenian territories and long-term genetic stasis followed by recent differentiation in different stable habitats. Furthermore, the star-like arrangement of plastid DNA haplotypes (Fig. 1) and DIYABC models suggest a pattern of long term survival and *in situ* differentiation. These results strongly agree with the idea of an ancient haplotype (I) widespread throughout the Tyrrhenian islands where the plant is present today, with different geographically scattered younger *in situ* derived haplotypes. In most cases, they represent endemic local variants that originated in isolation from each other, probably due to insularity or geography, on the one hand, and to the scattered availability of rupicolous habitats, on the other.

The Messinian Salinity Crisis, which has been cited to explain the distribution of many plant species in the Western Mediterranean (e.g., *Molins et al., 2011*) may also be invoked in this case, although the existence of Messinian terrestrial connections between the Corsica-Sardinia block and the Balearic Islands have never been documented (*Alvarez, 1972*; *Alvarez, Cocozza & Wezel, 1974*; *Rosenbaum, Lister & Duboz, 2002*). Also, although there is no evidence for further post-Messinian terrestrial connections between the major Tyrrhenian islands (*Alvarez, 1972*; *Alvarez, Cocozza & Wezel, 1974*; *Rosenbaum, Lister & Duboz, 2002*), direct land bridges existed during the Pleistocene glacial maxima between Corsica and Sardinia that allowed floristic exchanges (*Salvo et al., 2010*). This is also confirmed by the reconstruction of coastline during the LGM performed in this study (Fig. 1). The slightly exerted small capsules, and very small seeds (*López González, 1990*), and the plant's preference for shaded rocky sites (comophyte) are features that probably favoured short-distance dispersal. LDD of *A. balearica*, appears to be unfeasible during the Messinian when the Mediterranean Basin was a saline desert (*Hsü, 1972*). The fact that the plant lacks adaptations for over-water dispersal suggests also that LDD events between Majorca and the other Tyrrhenian islands (Corsica and/or Sardinia) were unlikely even during the Quaternary glacial maxima. No random LDD event was identified in the analyses performed in this study. Additionally, the star-like parsimony network inferred from plastid DNA data compiled (Fig. 1) is not consistent with a range-expansion model after LDD events, and no evidence was found for the existence of such events, either recent or ancient, between Majorca and the other Tyrrhenian islands derived from the almost nuclear AFLPs.

Historical gene flow seems to have existed between Corsican and Sardinian populations, as suggested by AFLPs. Both the NJ and PCoA analyses (Fig. 2) revealed no structuring of the overall genetic variability on a geographical basis. These results are also confirmed by the AMOVA analyses, which show that the genetic structure in four groups detected by NJ accounts for the comparatively highest amount of the total genetic variance, thus supporting the idea that only those populations from Majorca are to some extent genetically differentiated from the rest. The Bayesian analysis of population structure reveals active

historical gene flow and secondary contacts between Corsican and Sardinian populations (Fig. 2C). Particularly, clusters B and D are well represented on both islands but almost absent from Majorca (Fig. 2C) and the levels of admixture of these clusters tend to be higher among the populations located in southern Corsica and northern Sardinia (Fig. 2C). All these facts agree with the hypothesis of recurrent connections between Corsica and Sardinia in Miocene and Plio-Pleistocene times (Messinian Salinity Crisis: (*Gover, Meijer & Krijgsman, 2009*); Pleistocene glaciations: (*Lambeck et al., 2004*; *Lambeck & Purcell, 2005*)), which facilitated active exchanges of biota, as demonstrated for other organisms (*Zachos et al., 2003*; *Salvi et al., 2010*; *Fritz, Corti & Päckert, 2012*). By contrast, the plastid DNA data do not indicate significant post-Messinian floristic exchanges among Corsica, Sardinia, and the Tuscan Archipelago (only one haplotype is shared between Corsica and Sardinia), as proposed for other plant groups (e.g., *Quilichini, Debussche & Thompson, 2004*; *Salvo et al., 2008*; *Zecca et al., 2011*), a conclusion which may be biased by the fact that we were not able to establish good AFLP profiles for the plants collected in Montecristo and further highlights the importance of including anonymous hypervariable nuclear markers in phylogeographic studies.

## Evolutionary stasis and habitat stability in Mediterranean disjunct endemic taxa

The low levels of genetic variation found in the maternally inherited plastid DNA (i.e., low number both of detected and of missing haplotypes, low variation common to all the plastid DNA regions tested, and a maximum limit of four steps from the inferred ancestral haplotype were detected in the haplotype network) are consistent with some of the criteria that usually characterized palaeoendemic species (at least in the traditional broad concept of *Favarger & Contandriopoulos (1961)*. This low variation is usually interpreted as a consequence of long processes of adaptation in relative isolation to the intrinsic characteristics of the local refuge area (*Mansion et al., 2008*).

*Molins et al. (2011)* have emphasized that several relict endemic species show little or no morphological differentiation despite a long history of isolation on small continental fragments. Even though *A. balearica* was specifically cited in that work as an example of evolutionary stasis, this had never been demonstrated until now. The low mutation rates associated with the plastid genome in *A. balearica* probably correspond to low levels of genetic diversity detected also with AFLPs, thus revealing that stasis in this case agrees with generally low levels of genetic variation. A remarkable lack of variation in all plastid DNA markers scored (including intron regions, intergenic spacers, and plastid microsatellites) was detected for the Tertiary relict *Ramonda myconi* (L.) Rchb. (*Dubreuil, Riba & Mayol, 2008*), which was found to be in accord with previous results for other relict species (e.g., *Zelkova abelicea* (Lam.) Boiss. and *Z. sicula* Di Pasq., Garfì & Quézel by *Fineschi et al., 2002*; *Quercus suber* L. by *Magri et al., 2007*; *Cephalaria squamiflora* (Sieber) Greuter by *Rosselló et al., 2009*). According to *Dubreuil, Riba & Mayol (2008)*, the absence or low variation in the plastid genome could be a consequence of strong bottlenecks or genetic drift associated with small effective population sizes for maternally inherited markers (*Birky, Fuerst & Maruyama, 1989*), of slow population dynamics (*Dubreuil, Riba & Mayol,*

2008) and/or of slowed sequence evolution (*Dubreuil, Riba & Mayol, 2008*; *Molins et al., 2011*). The latter has been repeatedly associated with morphological stasis (*Barraclough & Savolainen, 2001*; *Soltis et al., 2002*; *Molins et al., 2011*). Nevertheless, *Casane & Laurenti (2013)* have recently suggested that, although a causal link between low molecular evolutionary rates and morphological stasis has been generally assumed, it seems that low intra-specific molecular diversity does not imply a low mutation rate, and also those intraspecific levels of molecular diversity and morphological divergence rates are under different constraints and are not necessarily correlated. As for *A. balearica*, independent markers suggest low levels of intraspecific molecular diversity (i.e., low plastid DNA variation, that seems to parallel the low overall genetic variability as revealed by a technique (AFLP) that covers the whole genome and also with low ITS sequence variation (J Bobo-Pinilla, J Peñas de Giles & MM Martínez-Ortega, 2013, unpublished data) that covers a small proportion of the nuclear DNA), but an explicit correlation between these data and either long-term morphological constancy or slowed mutation rates cannot be established with the available data.

Tertiary relict species have been forced to survive in refugia for long periods of time and their present genetic structure may therefore reflect the impact of a combination of ancient climatic and geographic changes. The ability to persist and resist overall adverse climatic conditions is probably coupled with the availability of relatively stable habitats, where intrinsic local properties have buffered the impact of historical climatic changes, thus allowing long-time persistence of particular species (*Thompson, 2005*; *Médail & Diadema, 2009*). The importance of local properties of refugia for survival of Tertiary relict taxa has previously been highlighted for other Mediterranean species, such as the rupicolous herb *R. myconi* (*Dubreuil, Riba & Mayol, 2008*). Furthermore, several authors (e.g., *Thompson, 2005*; *Peñas, Pérez-García & Mota, 2005*; *Rosselló et al., 2009*; *Youssef et al., 2010*; *Mayol et al., 2012*) have commented on the long-term stability of rupicolous habitats in the Mediterranean region and their role at warranting species survival based on the relatively low incidence of disturbances and interspecific competition and the fact that it is probably not fortuitous that many Mediterranean endemic species occur in rocky habitats (e.g., *Cymbalaria aequitriloba* (Viv.) A. Chev., *Nananthea perpusilla* DC., *Naufraga balearica* Constance & Cannon, *Soleirolia soleirolii* (Req.) Dandy, etc). *Arenaria balearica* represents a further example of the importance of rocky sites as conservation habitats and as long-term reservoirs of plant diversity within the Mediterranean region.

## ACKNOWLEDGEMENTS

We are grateful to Dr. Andreas Tribsch for providing samples for this study and for his initial help with lab work, Teresa Malvar for the lab work and D Nesbitt for the English-language revision.

### Funding

This work has been financed by the Spanish Ministerio de Ciencia e Innovación through the projects CGL2010-16357, CGL2009-07555, and CGL2012-32574. SBL was supported by a predoctoral research grant financed by AECID (Agencia Española de Cooperación Internacional para el Desarrollo). The funders had no role in study design, data collection and analysis, decision to publish, or preparation of the manuscript.

### Grant Disclosures

The following grant information was disclosed by the authors:
Spanish Ministerio de Ciencia e Innovación: CGL2010-16357, CGL2009-07555, CGL2012-32574.
AECID (Agencia Española de Cooperación Internacional para el Desarrollo).

### Competing Interests

The authors declare there are no competing interests.

### Author Contributions

- Javier Bobo-Pinilla performed the experiments, analyzed the data, contributed reagents/materials/analysis tools, wrote the paper, prepared figures and/or tables, reviewed drafts of the paper.
- Sara B. Barrios de León performed the experiments, analyzed the data, contributed reagents/materials/analysis tools, prepared figures and/or tables, reviewed drafts of the paper.
- Jaume Seguí Colomar, Giuseppe Fenu and Gianluigi Bacchetta contributed reagents/materials/analysis tools, reviewed drafts of the paper.
- Julio Peñas de Giles conceived and designed the experiments, contributed reagents/materials/analysis tools, wrote the paper, reviewed drafts of the paper.
- María Montserrat Martínez-Ortega conceived and designed the experiments, analyzed the data, contributed reagents/materials/analysis tools, wrote the paper, reviewed drafts of the paper.

### DNA Deposition

The following information was supplied regarding the deposition of DNA sequences:
GenBank sequences have been provided as a Supplemental File.

### Data Availability

The raw data has been supplied as Supplemental File.

### Supplemental Information

Supplemental information for this article can be found online at http://dx.doi.org/10.7717/peerj.2618#supplemental-information.

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
