# Peer review of "Phylogeography of Arenaria balearica L. (Caryophyllaceae): evolutionary history of a disjunct endemic from the Western Mediterranean continental islands"

_PeerJ, doi:10.7717/peerj.2618_

## Round 0.1 · original submission · Major Revisions

· Academic Editor

Major Revisions

Both reviewers have identified substantive problems with this manuscript. Of particular concern to me are (i) the small sample size for the plastid DNA analysis and (ii) problems with using Nested Clade Analysis. There are also many other serious criticisms that need to be addressed in a comprehensive revision.

Reviewer 1 ·

Basic reporting

The manuscript shows several sections (e.g. Introduction, Results, and Discussion) which are reiterative and made the text lengthy and repetitive in some parts. This particularly applies to the Introduction, which should be reduced and focused to the main point of the research.

Experimental design

This manuscript aims to assess the phylogeographic history of Arenaria balearica, an insular-distributed flowering plant from the Western Mediterranean. The authors use AFLP and plastidial markers to investigate the genetic structure and variability of this species. They conclude that the low levels of nuclear and plastidial DNA variation found reflect the evolutionary stasis of this species and shed light on the complex phylogeographic patterns within the Western Mediterranean basin. The manuscript fsll within the scope of the journal.
Unfortunately, the manuscript does not adequately explain the relevant conceptual questions to be addressed in this research. Yes, we know that the authors aim “to reconstruct the phylogeographic and evolutionary history of A. balearica” [lines 137-138]. But, (i) why this species was selected? (ii) why this species may a priori better contribute to the evolution study of the continental islands than other species?, and (iii) which research gaps, previously not assessed in other phylogeographic studies on the Western Mediterranean islands, were aimed to fill?
One of the main drawbacks of the research is the experimental design of the sampling. The use of a maximum of two individuals in each population for the plastid genotyping is unfortunate. This sampling is clearly inadequate and hardly be used as a baseline for assessing the genetic structure, variability, and phylogeographic history for this species. The authors should be aware of the larger sampling conducted in other studies they cited for assessing the infraspecific history of this Hercynian species. Currently, the plastid data constitutes no more than an initial pilot study. I strongly recommend the use, as a rule, of 10-15 individuals for each sampled population; lower samples did not agree with the high standard phylogeographic practices in use.
Several methods used to analyze the data are not free of controversy (e.g. Nested Clade Analysis), and their choice should be strictly discussed. In addition, the use of NCA with such a small sample size is risky and may lead wrong results.
Some terms and concepts used throughout the manuscript are misleading and should be clarified. For instance, “radiative evolution”, “disjunct distribution”, and “palaeoendemism” did not apply to the features showed by Arenaria balearica. Specifically, there is no evidence to support the status of this species as a palaeoendemism. I suggest touse the terminology given by Thompson (2005) in order to standardize its use. In addition, the authors should avoid “disjunct” and consider the use of “fragmented” as a better descriptor of the distribution of the species.
Lastly, I advice the authors to consider the term “homology” when dealing with AFLP markers. No evidence of such homology are given by the authors; co-migration in a electrophoretic system does not constitute any proof of homology.

Validity of the findings

Several findings reported in the manuscript are not robust (e.g. the average age of the cladogenetic event leading to A. balearica) and show a large confidence interval. This preclude any sound discussion about the timing of the evolutionary events involved in shaping the genetic structure of the species. Furthermore, there is not convincing explanations for the lack of a strong geographic structure of the obtained data. Lineage sorting has not been adequately discussed as well as the role of long range dispersion in shaping the genetic diversity found.

Reviewer 2 ·

Basic reporting

This is an interesting study about the evolution of Arenaria balearica, an endemic plant to Majorca, Corsica, Sardinia, and the Tuscan archipelago. The study uses a significant number of molecular techniques and methods to reliably infer phylogeographic patterns.
Although some results are convicing, there are quite a few number of points that should be addressed before providing a document to be published in Peer J.


Strong points
- AFLPs and plastid sequencing provide solid results about phylogeography at the population level.
- The Introduction clearly shows the interest of plant species distributed in Mediterranean islands to formulate explicit hypotheses.
- A relevant number of populations (29) and individuals (213) have been analysed.
- Methods are suitable for the characteristics of the datasets.
- Although there are some parts that need revision, English is good.


Weak points
There are conceptual problems along the manuscript; some related to an appropriate approach some to improve methods and result interpretation.

- Plastid vs. chloroplast: use always plastid when PCRing.
- What do you mean by radiative evolution? Is this a concept coined in phylogeography? If so, explain and include references.
- The Nested Clade Analysis is clearly unreliable (see below).
- The sister relationship between A. balearica and A. bertolonii is weak. The authors based their argument on a publication (Fori & Karis 2007). This publication is about Moehringia, where a few species of Arenaria (10 spp.) are also included. However, Arenaria is a genus that consists of over 150 species!
Besides, the low support found by Fori & Karis (2007) for A. balearica-A.bertolonii suggest that missing species are likely to be closely related (sister?) to A. balearica. This could change your results completely…

1. Title. It looks quite ambitious (“Disentangling the evolutionary history of the flora…”), when the authors only studied a single species.
2. Introduction. Quite long. Move sentences from line 118 to line 127 to M&M, to a new subsection (Study species).
3. Introduction. An explicit hypothesis and particular objectives are missing.
4. M&M. Values of the reproductive test are paramount to choose AFLP markers. Provide detailed information about this.
5. M&M. All information about calibration of your phylogeny (not nucleotide substitutions per site per year from another plants) is missing. Otherwise you divergence estimate is quite broad and useless.
6. M&M: the most important methodological flaw is the use of Nested Clade Analysis. This was deeply discussed in the past and no publications are based on this analysis anymore (see, for instance, Knowles 2008. Why does a method that fails continue to be used? Evolution 62: 2713-2717).
7. Results. You should also show results of haplotype networks not treating gaps as a fifth character. Bear in mind indels are often homoplasious.
8. Results. Your results are not clearly framed in terms of taxonomy and time scale along the ms. I recommend including all author´s data from phylogenies (for instance those based on ITS not provided) that support the results discussed in this study.
9. Discussion. As there is no explicit hypothesis, the dDscussion rambles.
10. Discussion. The authors claim for “long-term genetic stasis”. What do they mean for this? If you have genetic stasis, you will not have genetic variation and thus few genetic data for your analyses.
11. Discussion. There is a long discussion about LDD where authors neither provide experimental results about A. balearica dispersal nor discuss compelling studies about this topic.
12. Discussion. Is there any sound conclusion to be indicated?

Experimental design

This is an interesting study about the evolution of Arenaria balearica, an endemic plant to Majorca, Corsica, Sardinia, and the Tuscan archipelago. The study uses a significant number of molecular techniques and methods to reliably infer phylogeographic patterns.
Although some results are convicing, there are quite a few number of points that should be addressed before providing a document to be published in Peer J.


Strong points
- AFLPs and plastid sequencing provide solid results about phylogeography at the population level.
- The Introduction clearly shows the interest of plant species distributed in Mediterranean islands to formulate explicit hypotheses.
- A relevant number of populations (29) and individuals (213) have been analysed.
- Methods are suitable for the characteristics of the datasets.
- Although there are some parts that need revision, English is good.


Weak points
There are conceptual problems along the manuscript; some related to an appropriate approach some to improve methods and result interpretation.

- Plastid vs. chloroplast: use always plastid when PCRing.
- What do you mean by radiative evolution? Is this a concept coined in phylogeography? If so, explain and include references.
- The Nested Clade Analysis is clearly unreliable (see below).
- The sister relationship between A. balearica and A. bertolonii is weak. The authors based their argument on a publication (Fori & Karis 2007). This publication is about Moehringia, where a few species of Arenaria (10 spp.) are also included. However, Arenaria is a genus that consists of over 150 species!
Besides, the low support found by Fori & Karis (2007) for A. balearica-A.bertolonii suggest that missing species are likely to be closely related (sister?) to A. balearica. This could change your results completely…

1. Title. It looks quite ambitious (“Disentangling the evolutionary history of the flora…”), when the authors only studied a single species.
2. Introduction. Quite long. Move sentences from line 118 to line 127 to M&M, to a new subsection (Study species).
3. Introduction. An explicit hypothesis and particular objectives are missing.
4. M&M. Values of the reproductive test are paramount to choose AFLP markers. Provide detailed information about this.
5. M&M. All information about calibration of your phylogeny (not nucleotide substitutions per site per year from another plants) is missing. Otherwise you divergence estimate is quite broad and useless.
6. M&M: the most important methodological flaw is the use of Nested Clade Analysis. This was deeply discussed in the past and no publications are based on this analysis anymore (see, for instance, Knowles 2008. Why does a method that fails continue to be used? Evolution 62: 2713-2717).
7. Results. You should also show results of haplotype networks not treating gaps as a fifth character. Bear in mind indels are often homoplasious.
8. Results. Your results are not clearly framed in terms of taxonomy and time scale along the ms. I recommend including all author´s data from phylogenies (for instance those based on ITS not provided) that support the results discussed in this study.
9. Discussion. As there is no explicit hypothesis, the dDscussion rambles.
10. Discussion. The authors claim for “long-term genetic stasis”. What do they mean for this? If you have genetic stasis, you will not have genetic variation and thus few genetic data for your analyses.
11. Discussion. There is a long discussion about LDD where authors neither provide experimental results about A. balearica dispersal nor discuss compelling studies about this topic.
12. Discussion. Is there any sound conclusion to be indicated?

Validity of the findings

This is an interesting study about the evolution of Arenaria balearica, an endemic plant to Majorca, Corsica, Sardinia, and the Tuscan archipelago. The study uses a significant number of molecular techniques and methods to reliably infer phylogeographic patterns.
Although some results are convicing, there are quite a few number of points that should be addressed before providing a document to be published in Peer J.


Strong points
- AFLPs and plastid sequencing provide solid results about phylogeography at the population level.
- The Introduction clearly shows the interest of plant species distributed in Mediterranean islands to formulate explicit hypotheses.
- A relevant number of populations (29) and individuals (213) have been analysed.
- Methods are suitable for the characteristics of the datasets.
- Although there are some parts that need revision, English is good.


Weak points
There are conceptual problems along the manuscript; some related to an appropriate approach some to improve methods and result interpretation.

- Plastid vs. chloroplast: use always plastid when PCRing.
- What do you mean by radiative evolution? Is this a concept coined in phylogeography? If so, explain and include references.
- The Nested Clade Analysis is clearly unreliable (see below).
- The sister relationship between A. balearica and A. bertolonii is weak. The authors based their argument on a publication (Fori & Karis 2007). This publication is about Moehringia, where a few species of Arenaria (10 spp.) are also included. However, Arenaria is a genus that consists of over 150 species!
Besides, the low support found by Fori & Karis (2007) for A. balearica-A.bertolonii suggest that missing species are likely to be closely related (sister?) to A. balearica. This could change your results completely…

1. Title. It looks quite ambitious (“Disentangling the evolutionary history of the flora…”), when the authors only studied a single species.
2. Introduction. Quite long. Move sentences from line 118 to line 127 to M&M, to a new subsection (Study species).
3. Introduction. An explicit hypothesis and particular objectives are missing.
4. M&M. Values of the reproductive test are paramount to choose AFLP markers. Provide detailed information about this.
5. M&M. All information about calibration of your phylogeny (not nucleotide substitutions per site per year from another plants) is missing. Otherwise you divergence estimate is quite broad and useless.
6. M&M: the most important methodological flaw is the use of Nested Clade Analysis. This was deeply discussed in the past and no publications are based on this analysis anymore (see, for instance, Knowles 2008. Why does a method that fails continue to be used? Evolution 62: 2713-2717).
7. Results. You should also show results of haplotype networks not treating gaps as a fifth character. Bear in mind indels are often homoplasious.
8. Results. Your results are not clearly framed in terms of taxonomy and time scale along the ms. I recommend including all author´s data from phylogenies (for instance those based on ITS not provided) that support the results discussed in this study.
9. Discussion. As there is no explicit hypothesis, the dDscussion rambles.
10. Discussion. The authors claim for “long-term genetic stasis”. What do they mean for this? If you have genetic stasis, you will not have genetic variation and thus few genetic data for your analyses.
11. Discussion. There is a long discussion about LDD where authors neither provide experimental results about A. balearica dispersal nor discuss compelling studies about this topic.
12. Discussion. Is there any sound conclusion to be indicated?

Additional comments

This is an interesting study about the evolution of Arenaria balearica, an endemic plant to Majorca, Corsica, Sardinia, and the Tuscan archipelago. The study uses a significant number of molecular techniques and methods to reliably infer phylogeographic patterns.
Although some results are convicing, there are quite a few number of points that should be addressed before providing a document to be published in Peer J.


Strong points
- AFLPs and plastid sequencing provide solid results about phylogeography at the population level.
- The Introduction clearly shows the interest of plant species distributed in Mediterranean islands to formulate explicit hypotheses.
- A relevant number of populations (29) and individuals (213) have been analysed.
- Methods are suitable for the characteristics of the datasets.
- Although there are some parts that need revision, English is good.


Weak points
There are conceptual problems along the manuscript; some related to an appropriate approach some to improve methods and result interpretation.

- Plastid vs. chloroplast: use always plastid when PCRing.
- What do you mean by radiative evolution? Is this a concept coined in phylogeography? If so, explain and include references.
- The Nested Clade Analysis is clearly unreliable (see below).
- The sister relationship between A. balearica and A. bertolonii is weak. The authors based their argument on a publication (Fori & Karis 2007). This publication is about Moehringia, where a few species of Arenaria (10 spp.) are also included. However, Arenaria is a genus that consists of over 150 species!
Besides, the low support found by Fori & Karis (2007) for A. balearica-A.bertolonii suggest that missing species are likely to be closely related (sister?) to A. balearica. This could change your results completely…

1. Title. It looks quite ambitious (“Disentangling the evolutionary history of the flora…”), when the authors only studied a single species.
2. Introduction. Quite long. Move sentences from line 118 to line 127 to M&M, to a new subsection (Study species).
3. Introduction. An explicit hypothesis and particular objectives are missing.
4. M&M. Values of the reproductive test are paramount to choose AFLP markers. Provide detailed information about this.
5. M&M. All information about calibration of your phylogeny (not nucleotide substitutions per site per year from another plants) is missing. Otherwise you divergence estimate is quite broad and useless.
6. M&M: the most important methodological flaw is the use of Nested Clade Analysis. This was deeply discussed in the past and no publications are based on this analysis anymore (see, for instance, Knowles 2008. Why does a method that fails continue to be used? Evolution 62: 2713-2717).
7. Results. You should also show results of haplotype networks not treating gaps as a fifth character. Bear in mind indels are often homoplasious.
8. Results. Your results are not clearly framed in terms of taxonomy and time scale along the ms. I recommend including all author´s data from phylogenies (for instance those based on ITS not provided) that support the results discussed in this study.
9. Discussion. As there is no explicit hypothesis, the dDscussion rambles.
10. Discussion. The authors claim for “long-term genetic stasis”. What do they mean for this? If you have genetic stasis, you will not have genetic variation and thus few genetic data for your analyses.
11. Discussion. There is a long discussion about LDD where authors neither provide experimental results about A. balearica dispersal nor discuss compelling studies about this topic.
12. Discussion. Is there any sound conclusion to be indicated?

---

## Round 0.2 · accepted · Accept

· Academic Editor

Accept

I am impressed by the additional work done to address the reviewer's concerns. I have made some corrections to the language in the Introduction and Discussion sections. These appear in the uploaded pdf. Please check the language in the other sections.